# Electrochemical Aptasensors Based on Hybrid Metal-Organic Frameworks

**DOI:** 10.3390/s20236963

**Published:** 2020-12-05

**Authors:** Gennady Evtugyn, Svetlana Belyakova, Anna Porfireva, Tibor Hianik

**Affiliations:** 1A.M. Butlerov’ Chemistry Institute of Kazan Federal University, 18 Kremlevskaya Street, 420008 Kazan, Russia; Svetlana.Belyakova@kpfu.ru (S.B.); Anna.Porfireva@kpfu.ru (A.P.); 2Analytical Chemistry Department of Chemical Technology Institute of Ural Federal University, 19 Mira Street, 620002 Ekaterinburg, Russia; 3Department of Nuclear Physics and Biophysics, Comenius University, Mlynska dolina F1, 842 48 Bratislava, Slovakia

**Keywords:** electrochemical biosensor, aptasensor, metal-organic frameworks, 3-D networks, reticular materials

## Abstract

Metal-organic frameworks (MOFs) offer a unique variety of properties and morphology of the structure that make it possible to extend the performance of existing and design new electrochemical biosensors. High porosity, variable size and morphology, compatibility with common components of electrochemical sensors, and easy combination with bioreceptors make MOFs very attractive for application in the assembly of electrochemical aptasensors. In this review, the progress in the synthesis and application of the MOFs in electrochemical aptasensors are considered with an emphasis on the role of the MOF materials in aptamer immobilization and signal generation. The literature information of the use of MOFs in electrochemical aptasensors is classified in accordance with the nature and role of MOFs and a signal mode. In conclusion, future trends in the application of MOFs in electrochemical aptasensors are briefly discussed.

## 1. Introduction

Aptasensors are biosensors containing aptamers as biorecognition elements [1,2]. Their progress started since 1990, where several papers reported the method of aptamer selection and emphasized advantages of these new molecules over antibodies [3,4,5]. The term aptamer was derived from Latin *aptus* (to fit) and Greek *meros* (part) [6]. The aptamers are synthesized in vitro from a random library of nucleotides followed by their selection by means of affine chromatography containing the conjugates of the analyte molecules. The protocol called SELES (Systematic Evolution of Ligands by EXponential Enrichment) [7] consists of several cycles of selection and amplification of the number of copies that allows specification of the sequences with maximal affinity toward the target [8]. To some extent, the reaction of an aptamer with an analyte is similar to the antigen–antibody interactions and results in reversible formation of a molecular complex. In some cases, such complexes can exert enzyme-like catalytic activity. Thus, the aptamer-hemin complex mimics peroxidase activity [9]. Such complexes have found analytical applications in the assembly of appropriate biosensors [10].

Aptamers are more stable in comparison with proteins in extreme conditions, including the interference with proteases and microbial digestion [11]. They can be easily modified by some tags and functional groups intended for their implementation in the assembly of biosensors by covalent binding [12,13] or to the attachment of labels and linkers [14]. RNA aptamers have found predominant application in fluorescent protocols of signal measurement [15] though DNA aptamers in the assembly of electrochemical biosensors [16]. The number of analytes determined with aptamers has enormously increased during the past decades. Aptamers are successfully applied for the determination of proteins [17,18], drugs [19,20], cancer biomarkers [21,22,23], living cells (pathogenic bacteria, circulating cancer cells) [24,25,26], etc. For monitoring of the aptamer–analyte interactions, various optical techniques have been used, mostly derived from fluorescence [27,28] and absorbance [29,30,31] spectroscopy. Meanwhile, electrochemical transduces remain the most frequently applied in the assembly of aptasensors due to the obvious advantages they possess [32,33,34,35]. Such advantages involve a rather simple and reliable design of the instrumentation, well-elaborated theory of the signal, high sensitivity of detection, and compatibility with biochemical components, including aptasensors and their conjugates with auxiliary components. It is also important that electrochemical techniques are suitable for portable biosensor design in the framework of the point-of-care diagnostics concept. Many of the solutions in the electrochemical aptasensors follow from those previously elaborated and tested in appropriate immuno- and DNA sensors [36]. Universal design makes the implementation of electrochemical aptasensors in existing systems of medical diagnostics easier.

However, the number of analytes determined with electrochemical aptasensors remains lower than that of immunosensors. The problems related to the electrochemical wiring of the aptamers to the electrode are most often mentioned as reasons limiting further progress in the aptasensor design and application. Being electrochemically inactive, aptamers and aptamer-analyte complexes worsen the conditions of the electron exchange on the aptasensor interface and partially depreciate amplification of the redox response caused by mediated electron transfer or electrocatalysis. This calls for the elaboration of novel approaches to the signal amplification that consider steric limitations of the target interactions and their prevailing effect on redox conditions and performance of aptasensor.

Nanomaterials, consisting of redox-active or electroconductive materials like noble metals [37], carbonaceous particles (carbon nanotubes [38], graphene-related materials [39,40,41], carbon dots [42]), offer excellent opportunities for biosensor development. The diversity of their properties, wide variety of modification protocols, participation in the electron transfer, and extension of the active surface area make it possible to sufficiently increase the sensitivity of appropriate electrochemical sensors. Meanwhile, the selectivity of the response of biosensors is mainly affected by biochemical components. Among many other factors, non-specific interactions and adsorption on the biosensor interface limit the sensitivity and selectivity of the aptasensors. The contribution of such non-specific interactions is also growing with the surface area. For this reason, conventional nanoscale modifiers cannot guarantee reliable detection of biomolecules present in rather complex media at very low levels.

The complications mentioned above call for a further search of the materials combining high efficiency of the signal transduction and selectivity of the response. Coordination polymers, or coordination networks, consist of an inorganic part, mostly metal atoms, their clusters or metal oxides, and organic linkers. In 1999, the acronym ‘‘MOF” (metal-organic framework) was proposed for such systems by the Yaghi group [43]. Contrary to other similar composites, MOFs have a crystalline structure and assume the existence of some voids suitable for filling with solvents and analyte molecules.

Various MOF structures are easily obtained by solvothermal methods. They can exert mediator properties due to ability for reversible electron exchange by means of transient metal ion/clusters involved in their structure and high porosity and ability of selective discrimination of small molecules by their size, charge, and lipophilicity. Recently, several comprehensive reviews have been published to summarize the experience in the application of MOFs in electrochemical sensors and biosensors [44,45,46,47,48,49]. They consider various aspects related to the diversity of biochemicals introduced in the surface layer and electrochemical properties of the MOFs themselves and hybrid materials implementing MOF particles.

In this review, main attention is paid to the aptasensors with the MOF particles in their assemblies intended to improve performance in the determination of various analytes demanded in medicine, pharmacology, and environmental monitoring.

## 2. General Characterizations of MOFs

In accordance with the IUPAC definition [50], MOF is “a coordination network with organic ligands containing potential voids”. In turn, a coordination network is described as “a coordination compound extending, through repeating coordination entities, in 1 dimension, but with cross-links between two or more individual chains, loops, or spiro-links, or a coordination compound extending through repeating coordination entities in 2 or 3 dimensions”. The term previously used for such structures (hybrid organo-inorganic materials) is considered as imprecise because about all the coordination compounds can be ascribed with this wording. Although the above definition does not assume a crystalline character of the MOF structure, it seems important to distinguish them from the more general term “coordination polymer”. In addition to the above terms, some other reticular materials with the properties similar to those of the MOFs are known from the literature. They involve zeolitic imidazolate frameworks [51] and covalent organic frameworks (COFs) [52]. In the latter case, organic building blocks create 3-D networks on themselves, whereas MOFs always include bridging metals.

From the point of view of assembling from appropriate precursors [53], the MOF structures consist of the metal atoms or metal clusters (secondary building blocks) linked with polytopic ligands like terephtalic acid or imidazole fragments. The structures of the most common linkers applied are presented in Figure 1.

Metal clusters commonly have a polyhedral structure. Depending on the linkers, they can have various coordination numbers. Thus, Zr MOFs with coordination numbers of 6, 8, 10, and 12 have been reported [54]. This determines the network topology and formation of unusual particle forms like cubic, hexagonal particles, etc. [55].

Most common MOFs described in the assembly of electrochemical sensors and especially biosensors are constructed from carboxylate-based linkers. Their stability toward hydrolysis depends on the strength of the bonds between the carboxylate groups and the metal-based nodes. From the hard/soft acid/base principle, hard carboxylates form more robust structures with the metal ions in a high valence state (Ce(III/IV), Zr(IV), Hf(IV), etc.) [56,57,58,59]. MOFs assembled from Ni-, Cu-, and Co-based nodes and carboxylate-bearing linkers degrade to appropriate oxides in basic media [59]. For the same reason, such MOFs can be used for the synthesis of derived materials with electrocatalytic properties. Among the MOFs based on other metals, many reported examples refer to UiO-66 consisting of hexa-zirconium nodes and linear dicarboxylate linkers [60,61]. They exert high chemical stability both in acidic and neutral media but are redox inactive and hence should be first labeled with some appropriate groups like porphyrin fragments [62]. In aptasensors, the high affinity of Zr oxides toward phosphate groups in the DNA sequence make such materials attractive in the immobilization of aptamers with no respect of the protocol of signal measurement. MOFs based on nitrogen-containing linkers are considered as an alternative to the described carboxylate-based MOFs. The ZIF (zeolitic imidazolate framework) family is most popular in electrochemical detection systems. They are assembled with Zn^2+^ and Co^2+^ cations and are quite stable even in basic media due to the formation of rather narrow and hydrophobic pores [63].

In some cases, metal ions form with appropriate linkers the 2-D structures (linear supramolecular polymers), which are then assembled in the 3-D networks with participation of another metal ion. Such bimetallic MOFs show a uniform regular structure. Their properties, e.g., redox activity, porosity, internal volume, and hydrolytic stability, depend on both implemented metals. Ni cations are often applied as bridging ions. They provide assembly of the hierarchical structure, whereas Zn can be used for primary complexation with organic linkers [64]. In such materials, the integer ratio of the metal atoms (1:1, 1:2, 1:3) is mostly observed and the properties of the products highly depend on the ratio of the reagents mixed in the solvothermal synthesis.

Alternatively, bimetallic heterostructured MOFs can consist of two separated materials (MOFs-on-MOFs) or result from consecutive assembling of appropriate layers (core-shell, or MOFs@MOFs structures) [65]. Such heterostructured materials attract attention due to the higher variety of the properties of the materials and the synergetic effect of both components in the reactions of electrocatalysis and analyte recognition. Combination of two metals offers higher hydrolytic stability of the products against mechanical mixture of appropriated MOFs taken separately [66].

The morphology of common MOF materials is outlined in Figure 2. The 2-D films consisting of linear supramolecular polymer chains (Figure 2A) are mostly assembled on solid supports, e.g., electrode surface or crystalline carrier with regular alternation of the binding sites. Figure 2B represents the MOF crystal based on a single metal and polydentant ligand participating in formation of the chain linking bonds. Figure 2C shows a bimetallic MOF particle, where a second binding metal atom forms the 3-D structure of the product. Finally, Figure 2D,E outline heterogeneous MOF-on-MOF and MOF@MOF (core-shell [67]) structures, respectively.

In general, about 70,000 crystalline structures, which exert unique topology, structural backbone, and appended functional groups, have been reported in the Cambridge Structural Database to 2017 [68].

The rather simple protocol of assembly anfogurd variety of the building blocks and linkers make it possible to customize the structures with a wide range of pore size distribution. As a result, the internal surface area of the MOF structures can reach 8000 m^2^/g against 1000 m^2^/g for zeolites and 3500 m^2^/g for activated carbon [69]. For this reason, MOFs have found increasing application in gas storage and separation [70,71,72], drug delivery [73], and as supporting matrix for catalysts [74]. Recently, similar structures have become popular in sensors and biosensors as carriers of recognition elements and signal-generating agents. It should be noted that the high porosity of the MOF crystals is not the only requirement for their successful application in the analyte accumulation and recognition. In many cases, release of the molecules from the internal volume of the MOF pores results in disturbance of their regular structure followed by decay of the adsorption capacity and specificity toward the species determined. This prevents sensor/biosensor recovery after the contact with an analyte and their multiple use. The problem can be solved by formation of the hierarchical structures described above. Thus, Ru bipyridine complexes suitable for electrochemiluminescence measurements and able to form open frameworks are mostly used as modifiers in more rigid structures based on carboxylate-bearing linkers [75,76]. Another approach suggests the combination of the Ru complexes with bridging metal centers [77].

Pore tuning is considered as one of the principal advantages of the MOF structures in chemical analysis and sensor assembling. Thus, the MOF-5 structure assembled from Zn acetate and terephtalic acid possesses a 3-D cubic network with 12 Å pores. Substitution of the linker with 2-metyhlimidazole changes the topology of the network to that typical for zeolites with large open pores of 11.6 Å in diameter interconnected with small ones (3.4 Å). Similar MOF with Cr_3_O clusters has a cubic structure with extra-large pores (30 to 34 Å) [78]. Thus, variation of the precursors allows the assembly of various structures that are uniform on the atomic level and tunable in pore size and its variation.

Another way to modify the 3-D structure of the MOFs is to specify synthesis conditions, namely, to use surfactants and stabilizers like polyvinylpyrrolidone (PVP) [79] or poly(ethylene imine) (PEI) [80] added on the stage of the precursors’ mixing and dispersion of low-soluble components in water or aqueous solvents. Thus, the synthesis of Ni MOF with 2-aminoterephtalic acid as a linker has been described in dimethylformamide (DMF), its mixture with water, and that containing PVP [81]. In pure organic solvent, spindle-shaped particles of 1−1.5 μm were obtained, and in 50% DMF, the size of the particles decreased to 700−800 nm and their shape changed to irregular spheres. The addition of the surfactant decreased the size of spherical particles to 350−400 nm. Their shape became more regular. The voltammetric response to methylene blue as a redox indicator changed with the size of the MOF particles deposited on the electrode so that smaller particles produced a higher current. The influence of the solvent was attributed to the deprotonation of carboxylate groups of the linker promoted by water addition and to the PVP influence on the rate of the MOF nuclei growth. Together with the temperature of solvothermal synthesis, the choice of solvent or their mixtures makes it possible to affect the morphology of the particles obtained and their electrochemical properties. However, changes in the currents recorded can be related to the difference in the specific surface area of the particles or in the surface-to-volume ratio if intrinsic redox properties of the MOFs are measured. The influence of the shape itself is less mentioned and discussed.

Three types of the MOFs in the assembly of the aptasensors are described:“Pure MOFs” obtained from the metal salt and organic linker providing assembly of the 3-D structure. They are mostly synthesized in rather mild conditions (at 80−140 °C) in an autoclave or even at atmospheric pressure during 40−240 min [82]. The product is precipitated, separated, and washed from residuals of reactants and the solvents. Then, the particles are dried and used as received without further mechanical disintegration. Such MOFs mostly contain transient metals (their clusters), which can be involved in redox reactions or act as adsorbents for accumulation of the redox indicators and aptamers. Bimetallic MOFs of the regular structure (MOF-on-MOF and MOF@MOF) described above can also be included in this group though their synthesis is complicated and involves separate steps of the reagent’s addition.Hybrid materials consisting of the MOF as a platform for deposition of other components accumulated in their internal volume (low molecular species like redox-active dyes [83]) or on the surface. From inorganic species, hybrid particles can contain metal complexes [84] and oxides [85]. They can vary by the degree of surface loading, conditions of modification, and the properties of the products obtained. Organic modifiers can also be covalently attached to the functional groups of organic linkers (amino group of 2-aminoterephtalic acid as an example). Modification can simplify the following attachment of the MOF-based particles in the biosensor assembly or exert amplified redox properties attributed to the modifier.Nanoparticles obtained by calcination of the MOFs prepared in advance by conventional methods of low-temperature solvothermal synthesis. Actually, the product of calcination does not belong to the MOF family because high temperatures result in carbonization of organic matter and seriously influence the initial 3-D structure of the particles. Nevertheless, the MOF-derived materials show attractive electroconductive and electrocatalytic properties due to the high concentration of mesoporous carbon and oxides of transient metals able to complete redox reactions [86]. Fe_3_O_4_-modified particles derived from Fe containing MOFs in calcination are also used in magnetic separation as a part of the measurement protocol [87].

MOFs are commonly characterized prior to use in the biosensor assembly using conventional tools of nanotechnologies and nanomaterials investigation. Thus, the structure and elemental content of the particles are confirmed by attenuated total reflectance FT-IR spectra, X-ray diffraction analysis and energy-dispersive X-ray spectroscopy (EDX elemental mapping), the morphology and size of the particles by scanning electron microscopy (SEM) and transmission electron microscopy (TEM), surface charge, and hydrodynamic diameter by dynamic light scattering (DLS) [66].

In many cases, the characterization of the MOF particles is performed for a certain content of the reactant mixture used for the synthesis together with the results of molecular modelling and crystallographic presentation of the MOF unit structure (see Figure 3 as an example). This might be useful to follow the relationships between the chemical content and structure of the MOF particles. However, it seems insufficient to assess the contribution of the MOFs to the sensor/biosensor performance. Moreover, such characterization is mostly related to idealized conditions and does not ascribe the same MOFs in contact with the liquid phase, especially on the electrode–solution interface. For the same reason, an unusual shape of the MOF particles quite different from that of more common metals sometimes looks amazing but does not influence the electrochemical properties of the MOF material, mostly independent of the particular crystallographic description.

Adsorption capacity, internal pore volume, and other porosity characteristics are described by nitrogen adsorption isotherm and differential scanning calorimetry–thermoporosimetry [66]. It should be noticed that the characteristics determined by these methods have a limited relation to the performance of electrochemical sensors. Hydration and specific interactions with polar water molecules and their adsorption on internal walls of the pores significantly affect the quantities of the reagents that can be adsorbed in the MOF particles implemented in the sensor. For this reason, electrochemical aptasensors are mostly characterized by the signals of redox species adsorbed.

In addition, electrochemical impedance spectroscopy (EIS) provides valuable information on the conditions of electron exchange on the electrode interface and offers broad opportunities for monitoring various steps of the assembly of the surface layer [89]. In the EIS experiments, the ferri/ferrocyanide redox pair [Fe(CN)_6_]^3−/4−^ is used at its equilibrium potential. Changes in the electron transfer resistance and to a lesser extent the interface capacitance make it possible to conclude about the charge of the layer and its permeability for small ions. It is important that the redox indicator has the negative charge like phosphate residuals of the aptamer chain. This excludes interferences caused by non-specific adsorption or electrostatic accumulation of the redox indicator.

## 3. Aptasensor Assembling and Operation

### 3.1. General Assessment and Bibliography Statistics

Biosensor assembling is a protocol of consecutive attachment of biochemical and auxiliary components on the transducer interface to reach optimal conditions for target biochemical interactions and signal generation. Regarding electrochemical biosensors, heterogeneous mediators of electron transfer are mostly required to compensate for decreased conductivity of the surface layer caused by implementation of biopolymers and to amplify the signal of redox active labels and indicators. Immobilization of biomolecules should maintain their microenvironment to be most comfortable for biochemical functioning and protect them from reactive species and undesired working conditions (extreme pH, influence of reactive species on the steps of biosensor assembly and signal measurement, etc.). Besides, immobilization should take into account steric limitations of the analyte binding related to the partial shielding of biochemical receptors on the electrode interface. In the history of biosensors, many approaches have been elaborated for biosensor assembly that offer stable signal and long-term operation of biosensors in different and changing conditions. Below, some of them are briefly considered in relation to the use of the MOFs in aptasensors. It should be clearly noted that the protocols described below do not exhaust all the methods and are limited with those most frequently used within the subject of the review.

Although MOFs belong to rather new materials that appeared in the scope of chemists in the past few decades, they have found intensive application in electrochemical sensors and biosensors from the very beginning. Figure 4 represents the results of the bibliography statistics performed with Web of Science to follow annual changes in the number of publications and general principles of the MOF application in the aptasensor assembly.

Starting from 2012, an exponential growth of the articles devoted to the use of electrochemistry tools in MOF synthesis and application was observed. Meanwhile, a significant part of such works utilized redox activity of the metals in the assembly of the MOF particles as a way to assess the structural specificity of the materials and to monitor the assembling of reticular structures from precursors, especially in 2-D nets on solid supports. Then, the electrocatalytic properties of the metal clusters in the knots of the frameworks were successfully applied in the protocols of mediated oxidation of many organic species attractive from the point of view of medical diagnostics or food safety control. Application of the MOFs in aptasensors covers about 10−12% of the publications and among them about half are devoted to electrochemical aptasensors. As could be seen from the histograms, enormous growth of the interest in the MOF-based aptasensors is expected in the nearest future. It is related to the remarkable variety of their properties and obvious advantages briefly mentioned above in the introduction.

The role of the MOFs in the aptasensors described (Figure 4D) varies from a rather traditional application as a surfaced enhancer and to their use as a source of redox mediators collected or released in the target binding event. MOFs are also compatible with biochemical strategies of the signal amplification assuming the use of enzymatic and DNAzyme amplification or exonuclease-assisted DNA amplification protocols. To some extent, the sandwich assay with two aptamer/DNA sequences utilized for immobilization of the bioreceptor and its labeling can be considered as a biochemical approach previously elaborated for immunoassay. All of the assemblies combining MOFs and aptamers with electrochemical transducer are considered in more detail below in the review.

### 3.2. Assembling of the Surface Layer

As was briefly mentioned above, MOF particles are mostly deposited on the surface of the transducer by drop casting. In some cases, the electrode can serve as a solid support for their synthesis if it does not assume necessity in high temperatures and pressure. In such cases, the nature of the electrode material is important because it determines the structure of the modifier and characteristics of the electron transfer. Contrary to most modern biosensors, aptasensors with the MOF particles are mostly designed on gold instead of glassy carbon electrodes (GCEs) or screen-printed carbon electrodes frequently applied in enzyme or immunosensors. This might be due to the compatibility of gold with the protocol of the aptamer immobilization via formation of the Au-S bonds between the electrode and terminal thiol group of the aptamer molecule. In addition, Au nanoparticles can be chemically or electrochemically deposited onto the electrodes or on the MOF particles to increase their specific surface area.

Deposition of the MOFs can be accompanied with the introduction of other nanomaterials. Besides the Au nanoparticles mentioned, carbonaceous nanomaterials play the same role of surface enhancement and increase the quantities of aptamer molecules attached to the surface. Carbon nanotubes, reduced graphene oxide, and g-C_3_N_4_ nanoparticles also increase the rate of electron transfer and serve as carriers for aptamer immobilization. Polymer binders like chitosan can extend the operation period of aptasensors due to the prevention of leakage of the surface components and better mechanical durability of the surface layer.

Aptamers are immobilized on the electrode interface by physical or covalent immobilization. In the first case, physical adsorption is mostly directed by electrostatic interactions [90]. Although pore volume and size are often discussed in the step of the MOF synthesis optimization, their importance for aptamer immobilization is questionable because of the rather large volume of bioreceptor molecules and necessity to provide accessibility of the analyte molecules, which are often rather big to fill in the internal volume of the MOF crystals. Electrostatic immobilization is addressed to the attraction of phosphate groups of the aptamer chain to the positively charged metal cations of the MOF units. In case of Zr MOFs, the covalent interaction of phosphate residues with ZrO clusters is also discussed as a driving force of the aptamer attachment [91].

Physical adsorption can be additionally amplified by polycationic species added on the stage of the MOF synthesis. The same approach is used for suspending MOF particles prior to their casting on the electrode surface [92]. Hydrophobic interactions of the aromatic π-electrons of the MOF linkers and of the nucleobases in the DNA main sequence are also mentioned as a driving force of physical adsorption [93].

In chemical (covalent) immobilization, terminal groups of aptamers form covalent bonds with certain functional groups of the electrode material or of the additives previously deposited on its surface. The previously mentioned interaction between thiolated aptamer and Au nanoparticles is also related to the covalent immobilization. In comparison with physical adsorption, covalent immobilization is less sensitive to the pH and reaction media content due to the higher stability of the covalent bonds toward hydrolysis and oxidation against coulomb interactions. Then, the formation of the bonds always involves a particular site of the electrode/carrier and of the aptamer molecules. This makes the structure of the product obtained predictable (so-called site-specific immobilization). In comparable conditions, covalently attached aptamer is more accessible for bulky analyte than that electrostatically immobilized. Sufficiently lower steric hindrance of target interactions of the aptamer-analyte is achieved by the introduction of long chain fragments (spacers) in the aptamer molecule. These spacers consist of methylene groups or homonucleobases poly(A)–poly(T) lift the aptamer over the electrode surface and simplify its interaction with the target molecule [94]. Spacer introduction increases the flexibility of the aptamer chain in its linear configuration and simplifies the conformational changes of the aptamer structure in the analyte recognition event. Spacers affect the hydrophobicity of the aptasensor interface and hence can influence the transfer of the organic analyte molecules from aqueous solution to the electrode surface for the following binding with the biorecognition element [95].

Covalent immobilization is mostly performed by bifunctional reagents that form bonds with the functional group in the assembly of the carrier or electrode material and with another one in the structure of an aptamer molecule. Aminated aptamers with terminal primary amino groups are commercially available and can interact with two such reagents, i.e., carbodiimides and glutaraldehyde. The reaction scheme is briefly outlined in Figure 5.

Glutaraldehyde reacts with many terminal functions but most efficiently with primary amino groups. In the latter case, the Schiff bases are formed. The reaction of the >C = N- group formation is reversible and sensitive to hydrolysis [96]. Additional treatment with a chemical reductant like NaBH_4_ results in the formation of a more stable amide bond. In the MOF particles, 2-aminoterephltalic acid can be involved in this reaction together with aminated aptamer.

Carbodiimide binding requires aminated and carboxylated reactants that interact with formation of the amido group. Carboxylic groups are present on the surface of carbonaceous materials like carbon nanotubes. Their number can be increased by preliminary oxidation of the carbon materials with strong oxidants. Carboxylic groups of the MOF linker in the structure of terephtalic acids etc. are hardly involved in such a reaction due to their participation in formation of the 3-D MOF structure. Among the many reagents described for carbodiimide binding, 1-ethyl-3-(3-dimethylaminopropyl)carbodiimide (EDC) and the cyclohexyl derivative are most often used for the aptasensor design. The reaction is accelerated by the addition of *N*-hydroxysuccinimide (NHS) [97]. Carbodiimide binding is fulfilled in mild conditions at ambient temperature or even at 4 °C. Similar processes can be used for the implementation of labels in the structure of aptamers and auxiliary DNA sequences used for the signal detection.

Affinity immobilization is based on the use of specific interactions with natural or artificial receptors that utilize or mimic natural biospecific interactions [98]. Avidin–biotin interaction is one of the most frequently mentioned examples. A biotin fragment is introduced in the aptamer structure (Figure 6) while a protein part (avidin or streptavidin) is placed on the electrode. For immobilization on gold, the synthetic derivative of avidin, neutravidin, which is from thiolated groups, is commercially available [99]. Each avidin molecule can bind up to four biotin fragments so that their interaction allows the formation of multilayered structures with participation of various biotinylated counterparts.

High affinity of avidin-biotin complexation (*K_D_* ~10^−14^–10^−15^ M [100]) makes this protocol competitive with covalent attachment of aptamer by carbodiimide binding. Affinity interactions involve other natural receptors specific to a certain fragment in immobilized molecules. Thus, concanavalin A is applied for protein binding via the glycan domain [101]. However, they are rarely used in aptasensor assembling.

Being structurally relative to conventional DNA probes, aptamers can be involved in hybridization with complementary DNA sequences immobilized on the electrode surface. Such an interaction can be also considered as affinity immobilization though double stranded helix formed can then dissociate in the presence of the analyte molecules and leave aptamer sequence to go away form the electrode interface (Figure 7).

Hybridization involves a rather short sequence of the aptamer and solves both the problems of immobilization and accomplishment of a proper orientation of the aptamer against the electrode surface. Interaction with bulky analyte is more effective with orthogonally standing DNA probe-aptamer duplexes. Auxiliary DNA applied for an aptamer collection is called capturing DNA contrary to the signaling DNA that makes the label detectable with an appropriate transducer. It is mostly immobilized to the electrode via Au–SH interactions [102,103]. Application of the capturing and signaling DNA sequences in the sandwich assay is described in more detail below.

### 3.3. Signal Measurement Protocols

Recently described aptasensors based on the MOF structures offer a great variety of approaches to the surface layer assembling and signal measurement. They influence the characteristics of appropriate analyte determination to a great extent. The performance of the MOF-based aptasensors is summarized in Table 1. The analytes are presented in alphabetical order. In the description of precursors, hydrate water is omitted from the metal salts formulae.

In electrochemical aptasensors, the following strategies of signal generation have been described and successfully used together with the MOFs:Measurement of permeability of the surface layer for small ions.Redox signals of labels implemented in the aptamer or auxiliary DNA sequences.Redox signals related to the MOF components (mainly transient metals).

#### 3.3.1. Permeability Assessment

These protocols are based on the fact that specific aptamer–analyte interactions result in the implementation of bulky inert molecules preventing the access of redox-active species to the electrode surface and hence decreasing their signal recorded by voltammetry. Alternatively, increased charge transfer resistance can be recorded in the presence of the [Fe (CN)_6_]^3−/4−^ ions as a redox probe. The scheme of the reactions is outlined in Figure 8.

This protocol first elaborated for immunoassay techniques shows good results for bulky analyte molecules like proteins or living cells. In case of small analytes, aptamers with guanine residues change their configuration from a linear one to a G4 quadruplex [150,151]. This results in a denser packing of the surface layer and suppresses the transfer of redox indicators to the electrode.

The MOFs are mostly utilized as aptamer carriers. The MOF particles can be decorated with Au nanoparticles to extend the surface area and simplify immobilization of thiolated aptamers. Non-specific adsorption is main factor influencing the sensitivity and selectivity of the response, especially in the presence of proteins attaching to the bare gold surface by cysteine residuals.

For this reason, the electrodes are additionally treated after the aptamer immobilization with inert organic thiols, preventing such adsorption. 6-Mercaptohexanol is used in many protocols due to its hydrophilicity and easy attachment to the naked part of the electrode. In both EIS and DPV measurements, a [Fe(CN)_6_]^3−/4−^ redox indicator is applied so that the sensitivity of both methods is expected to be close to each other. However, voltammetry often shows slightly lower limits of detection (LOD) than EIS contrary to the common behavior of similar immune- and DNA sensors.

On the one hand, this can be related to a high resolution of the peaks in DPV and the similar method of square wave voltammetry (SWV). On another hand, EIS data are much more affected by surface phenomena like counter ion transfer and high charge capacity of the interface in comparison with the voltammetry measurements. Meanwhile, the EIS parameters offer more information on the mechanism of signal generation. For this purpose, trends in the changes of the electron transfer resistance are compared with those of capacitance. If the diffusion factor dominates and increased resistance is caused by a lower rate of redox indicator transfer, capacitance is insensitive to the surface layer assembling, including the addition of an analyte that changes irregularly. If the charge transfer resistance is affected by the charge separation, the electron transfer resistance and capacitance tend to change in opposite directions (increasing resistance vs. decreasing capacitance). It should be mentioned that the impedimetric determination requires the selection of most suitable equivalent circuit for data fitting and control of non-ideal behavior of the system (correspondence of the constant phase element to the real capacitance of the electrode interface). The latter requirement related to the assessment of the roughness factor is rarely checked in the EIS measurements with the MOF-based aptasensors.

Another drawback of the measurement scheme described is the necessity to quantify the decay of the signal, which is maximal for a blank experiment and decreases with increasing analyte concentration. For such systems, not only the current (resistance) deviation but also the uncertainty of the concentrations related to the signal should be assessed, especially near the LOD value. The signal-to-noise ratio commonly used for the LOD calculation does not work properly for the calibration slopes with a relatively slow slope.

#### 3.3.2. Signals of the Redox Labels and Indicators

Although the [Fe(CN)_6_]^3−/4−^ indicator remains the most frequently used in aptasensors, other species have found increasing application for signal generation. Some of them denoted as labels are attached to the carrier or electrode surface while other ones (redox indicators) are diffusionally free and can be accumulated in the surface layer due to analyte binding. In case of MOFs, they can serve as specific sorbents for redox-active species. Their activity is then used for the signal measurement. Thionine [121,142,149], fullerene [126], ferrocene [121], and toluidine blue [128] have also been applied for voltammetric signal detection. The signal of labels covalently attached to the electrode surface was the most sensitive for the sandwich assay. In them, analyte is captured by the aptamer attached to the electrode and the complex formed is involved in specific binding with another (signaling) aptamer-bearing label. The scheme of the sandwich assay with aptasensor is schematically presented in Figure 9.

The sandwich assay is most effective when the MOF particles are utilized as labels. Thus, in the aptasensor for detection of β-amyloid oligomers as biomarkers of Alzheimer’ disease, both capturing and signaling aptamers were immobilized on the Au nanoparticles. However, capturing aptamer was attached to the GCE surface while signaling aptamer was placed on the surface of the Cu MOF. As a result of the sandwich assay, each analyte molecule resulted in the linkage of the MOFs particles to the electrode interface. Their quantities were monitored by the reduction peak current attributed to the partial reduction of Cu^2+^ ions in the assembly of the MOF particles [106]. It should be noted that the accessibility of the Cu^2+^ ions in the assembly of the MOF particles has not been proved by an appropriate electrochemical tool and the measurement required preliminary deaeration of the working solution to eliminate the interfering effect of dissolved oxygen. The Cu^2+^/Cu^+^ pair was also applied for detection of thrombin [139]. Here, Cu MOF based on 1,4-cyclohexanedicarboxylic acid was applied as a modifier of the GCE covered with the Au nanoparticles and thiolated aptamer. Specific interaction with thrombin decreased the DPV signal of Cu^2+^ though the reasons of such a response remain unclear.

The cathodic peak current of the Pb^2+^ ions released from appropriate MOFs treated with nitric acid was applied for the determination of carcinoembryonic antigen [109]. The sensitivity of the response was additionally increased by the hybrid chain reaction, which resulted in the formation of a dendritic DNA scaffold. The formation of appropriate products of the DNA amplification was monitored by a Ru(NH_3_)_6_^3+^ redox indicator. Its oxidation current was proportional to the quantity of the double-stranded DNA scaffolds on the electrode surface.

Zr MOFs synthesized from ZrCl_4_ and 2-aminoterephtalic acid and saturated with Pb^2+^ or Cd^2+^ ions were successfully applied for the determination of antibiotics [116,127]. The Zr MOFs particles were modified with appropriate aptamers and saturated with the metal ions. They were then immobilized by interaction with antibodies on the surface of magnetic beads. The reaction with target analytes resulted in release of the aptamers and their reaction with the drugs. Then, the products were digested with the RecJ_f_ exonuclease and analytes returned to the aptasensor for recycling. The released MOFs generated cathodic signals of Pb^2+^ and Cd^2+^ reduction. Their simultaneous determination by SWV made it possible to detect two drugs in one measurement. The reaction scheme was applied for sensitive determination of kanamycin, oxytetracycline, and chloroamphenicol. Similar to the above examples, Cd^2+^/Cd^0^ [129], Co^2+^/Co^3+^ [143], and Ce^3+^/Ce^4+^ [142,146] redox pairs derived from the MOF structure have been applied for the detection of other target aptamer–analyte interactions.

Methylene blue deserves special attention due to its importance and frequency of application in DNA sensors and aptasensors. It can intercalate double-stranded DNA helix and/or interact at minor grooves of the DNA helix. In addition, it can be covalently attached to the terminal tags of the aptamers and auxiliary DNA sequences and changes its redox signal in accordance with accumulation on the electrode interface or partial shielding in the target biomolecular interactions at the proximity of the electrode. In addition, methylene blue can be used for saturation of the MOFs utilized as a label of the aptamer immobilized by hybridization with the complementary DNA sequence (see above in Figure 7) [132]. The reaction with patulin as an analyte resulted in release of the aptamer from the surface so that the cathodic peak current of methylene blue was decreased. In a similar manner, aptamer bearing covalently attached methylene blue was first adsorbed on the black phosphorus particles mixed with Mn MOF on the electrode surface and then removed from the surface in specific binding of stress-induced phosphoprotein 1 [138] (Figure 10A). The reaction was monitored by a decrease of the DPV current of methylene blue reduction.

As an example of a more sophisticated approach of methylene blue application, a pinhole aptamer was covalently attached to the electrode and saturated with methylene blue accumulated in the hybridized part of the aptamer molecule (Figure 10B). Target reaction with *E. coli* results in opening of the circle of the aptamer and release of the methylene blue molecules so that its DPV signal decreased with the cell number in the sample tested [113].

#### 3.3.3. Other Detection Techniques

In addition to the high selectivity of aptamer–analyte interaction, the use of MOFs offers amplification of the signal via implementation of catalytic cycles. For this purpose, horseradish peroxidase [127,137] was used in the framework of the traditional competitive and sandwich assay with hydroquinone substrate and redox detection of benzoquinone formed in the reaction. Besides, catalytic reaction of H_2_O_2_ oxidation catalyzed by the MOFs bearing noble metals has been proposed for thrombospondin-1 [146] and led [133] determination. Pd nanoparticles on the Fe-based MOF carrier were involved in catalytic oxidation of tetramethylbenzidine monitored by voltammetry [122]. The measurement systems described can be considered as peroxidase-mimicking systems. The only example of other catalytic systems is the detection of lipopolysaccharides with the Ce-based MOFs bearing Au nanoparticles that catalyze oxidation of ascorbic acid [118]. In all the cases described, the aptasensors show LOD values below picomolar levels and a rather wide range of the analyte concentrations determined. The idea to use MOFs as a catalyst carrier is related to the rather high surface-to-volume ratio and significant increase of the catalyst quantities attached to the reaction site against conventional techniques with the same metal nanoparticles directly deposited on the electrode or introduced as labels in the reactants.

On the other hand, the possible interfering influence of readily oxidized compounds and partial poisoning of the catalytic particles by thiolated organic species can be expected. In most cases, the investigation of matrix effects in a real sample assay is rather limited and leaves space to think about possible underestimation of the analyte content in complex media.

## 4. Conclusions

MOFs belong to rather ‘new’ materials that appeared in the focus of the researchers involved in biosensor development about 10 years ago. Most of the literature devoted to the application of MOFs in electrochemical (bio)sensors covers the period from 2010. The interest in MOFs is mostly related to the diversity of their properties and was inspired to a high extent by great interest in nanotechnologies and nanomaterials in many areas of science and industry. It should be, however, noticed that most of the MOF materials described in the articles related to the design of the electrochemical sensors and biosensors do not belong to nano scale. Their size is mostly about 0.1-1 μm (mesoparticles). Hence, some obvious advantages of nanoparticles (high surface-to-volume ratio, catalytic activity, aggregation-disaggregation abilities, and high chemical and electrochemical activity) cannot be simply extended to the MOF materials and should be proved for each particular case. Application of the MOFs started from their use as reticular materials in gas sensing, storage, and separation. Traditionally, characterization of the new MOF materials is based on crystallography data and porosity assessment though aqueous solutions offer their own requirements and limitations for use of such particles. The idea to apply the MOFs as containers for small molecules (redox indicators) seems in this respect evident but not always effective. Thus, leaching metal cations from the internal volume of the MOF particles was mentioned as one of the reasons of the high background signal [98]. The use of metal atoms positioned in the knots of the 3-D networks requires further investigation because the rigid structure of the MOF can prevent direct contact of such cations with the electrode. To some extent, the last limitation can be excluded by implementation of the MOF particles into the conductive matrix [113,119,131]. In many cases, nanoparticles of noble metals are deposited on the MOF surface. They serve as support for the immobilization of thiolated aptamers and catalysts for voltammetric signal generation. In this sense, MOFs are hardly different from carbonaceous nanomaterials and advantages of their application do not seem obvious.

All of the above mentioned is applicable for electrochemical aptasensors utilizing MOFs to enhance surface square, immobilize aptamers, and generate redox signals. The use of bulky and electrochemically inactive aptamer molecules makes the aptasensor behavior sensitive toward the conditions of electron transfer. Contrary to electroconductive carbonaceous materials, MOF particles offer a shuttle mechanism of the electron exchange with participation of both free and immobilized mediators, including the metals as counterparts of the MOF structure. Such an electron transfer path is easily broken by the inclusion of analyte molecules. For this reason, aptamers based on the MOF structures often show a record sensitivity of detection especially for small analyte molecules that affect the transfer of conventional redox indicators like [Fe(CN)_6_]^3−/4−^ to a rather small extent if common approaches of the biosensor assembling are used.

Although most of the articles devoted to the electrochemical aptasensors based on MOF materials do not provide a comprehensive comparison of the characteristics with conventional analogs, it can be concluded that introduction of MOF particles can decrease the LOD values by at least one order of magnitude against the application of similar heterogeneous mediators of electron transfer. The difference might be even higher if MOF particles are used as parts of indicators attached to the electrode interface via target biochemical interactions with analytes. These benefits follow from the size of the particles and multiplication of the signal due to the much higher number of redox sites involved in the electron transfer. Appropriate examples are presented in Table 1. It should also be noted that other figures of merit like a dynamic range of concentration or measurement duration did not demonstrate similar progress against other biosensor materials. Then, application of biochemical approaches to the signal increase, including implementation of auxiliary pinhole aptamers and exonuclease-assisted DNA amplification, the level of influence of other components of the aptasensor assembly.

Summarizing the modern progress in the development of aptasensors based on the MOF materials, one can conclude that further achievements can be expected from the broadened application of hybrid MOF-on-MOF and MOF@MOF particles that combine the advantages of several metals and linkers and can be easily adapted to a particular measurement mode or signal detection requirements. Thus, the use of such hierarchic materials with Ru bipyridine complex shows high efficiency of electrochemilumenscent measurements [124]. Such hybrid metal particles are free from drawbacks related to the leaching metals from the modifier layer and offer better opportunities of the analyte measurements related to very low background signal. The sensitivity of the assay is greatly increased by covalent linking of the MOF particles as labels to the aptamer or auxiliary DNA sequences used.

The variety of hybrid particles can also be extended by variation of the organic linkers and conditions of the synthesis, which can involve chemical and electrochemical steps and variation of the solvents and temperatures applied. A similar conclusion can be made about MOF-derived materials, where the initially synthesized MOF particles are then calcinated at high temperature or treated with chemical reagents. The products do not formally belong to the MOF family but retain the porosity, specific shape, and high adsorption capacity typical for precursors. Although many such derived materials have an alternative path of synthesis, the use of MOF precursors might be beneficial from the point of view of simplicity, and labor and time intensiveness.

The following trends in the further progress of MOF-based aptasensors can be mentioned.

Faster progress is expected in the multiplex assay and simultaneous determination of several analytes based on the number of aptamers introduced in the aptasensor structure.Growing interest will be directed to the signal-on aptasensors, where an increased concentration of the analyte increases their response. This kind of sensor seems more beneficial both from the point of view of metrological assessment of the results and application of the aptasensors in the real sample assay.Although the term ‘MOF’ is mostly related to the use of the linkers with carboxylic binding groups, the variety of the organic part of the MOFs will be increased by the introduction of new structures and precursors derived from imidazole fragments and some others exerting an unusual architecture of the internal space of the particles and their morphology.The interest in the synthesis of new MOF materials will be shifted to their implementation in supramolecular structures consisting of supramolecular polymers and aptamer molecules.

In general, the high variability of the MOF structures and their easy accessibility as building blocks of aptasensors make it possible to rely on their broad application in medicine, food industry, and environmental pollution monitoring.

## Figures and Tables

**Figure 1 sensors-20-06963-f001:**
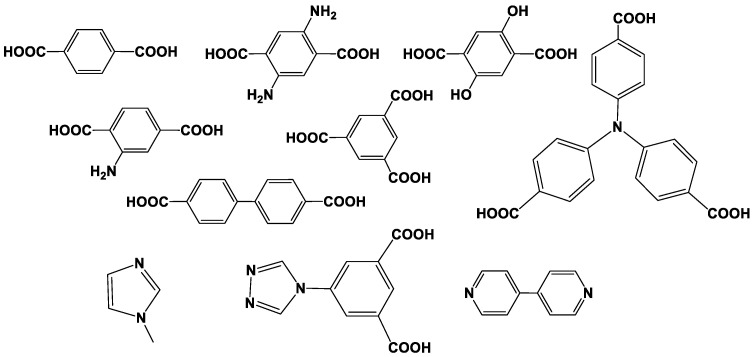
Chemical structures of the linkers applied for assembling the MOF materials.

**Figure 2 sensors-20-06963-f002:**
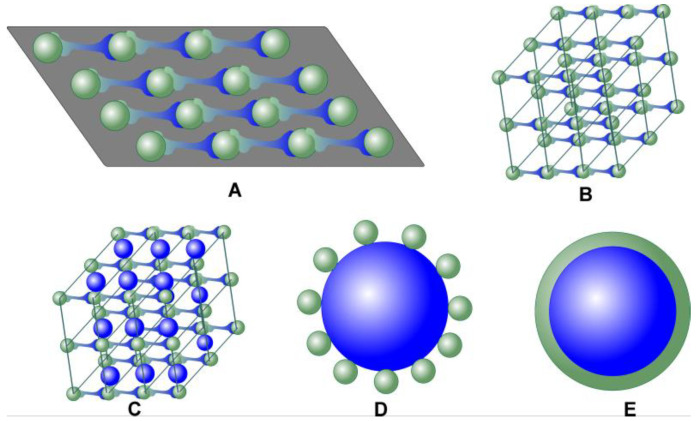
Schematic outline of various MOF structures. For a description, see the text above.

**Figure 3 sensors-20-06963-f003:**
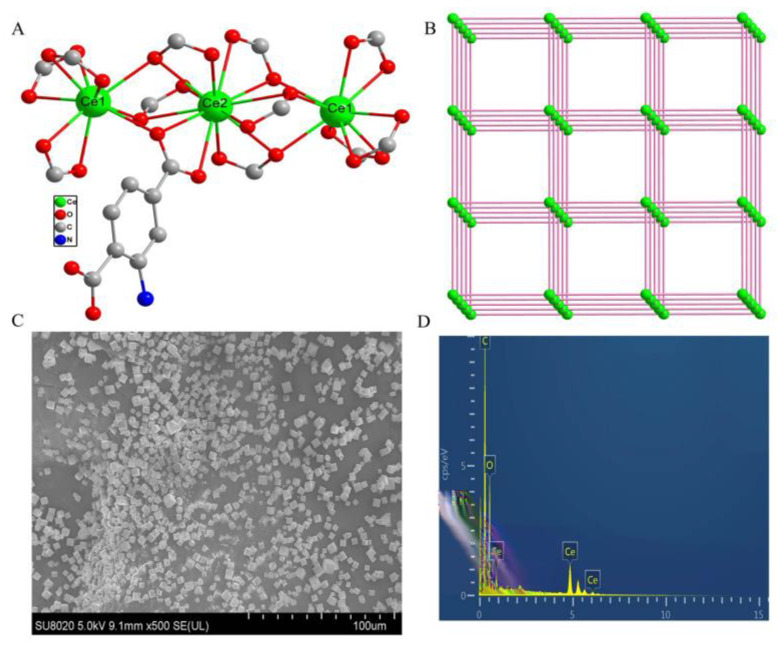
Characterization of the Ce-based MOF for the assembly of aptamer for ATP determination. (**A**) Coordination environments of Ce^3+^ ions and 2-aminoterephtalic acid; (**B**) the 3-D topological structure; (**C**) SEM image; (**D**) EDX elemental mapping [88].

**Figure 4 sensors-20-06963-f004:**
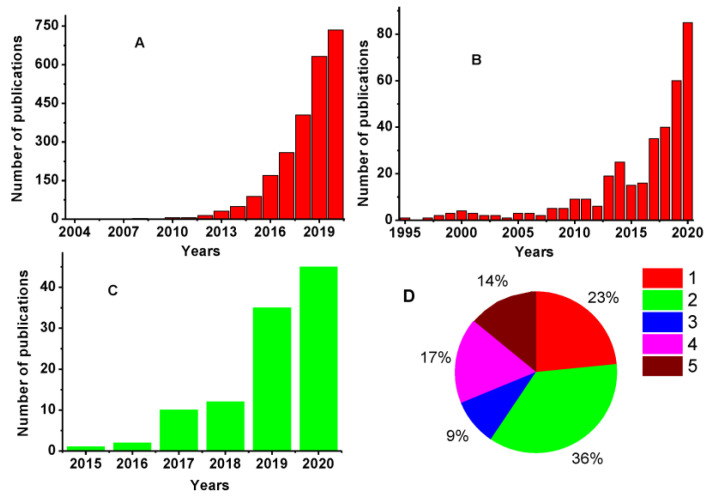
Analysis of the bibliography devoted to the application of aptamers in electrochemical sensors and biosensors performed with the Web of Science database. (**A**) ‘MOF’ and ‘electrochemical’ search; (**B**) ‘MOF’ and ‘aptasensor’ search; (**C**) ‘MOF’ and ‘electrochemical aptasensor’ search; (**D**) pie chart describing distribution of the signal measurement modes: 1—assessment of permeability of the surface layer; 2—application of biochemical amplification approaches; 3—sandwich assay; 4—measurement of intrinsic redox activity of the MOFs; 5—application of diffusionally free redox indicators.

**Figure 5 sensors-20-06963-f005:**
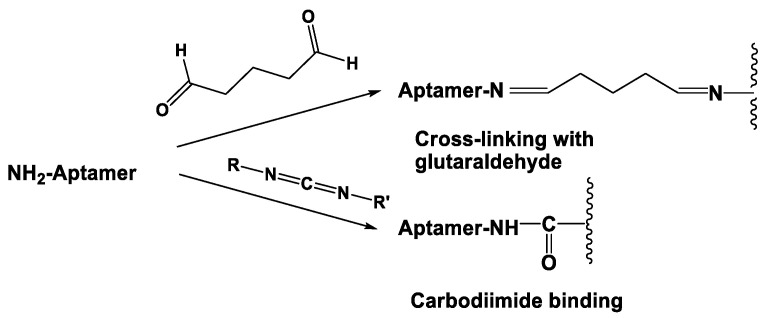
Covalent attachment of aminated aptamer to an insoluble carrier or electrode.

**Figure 6 sensors-20-06963-f006:**
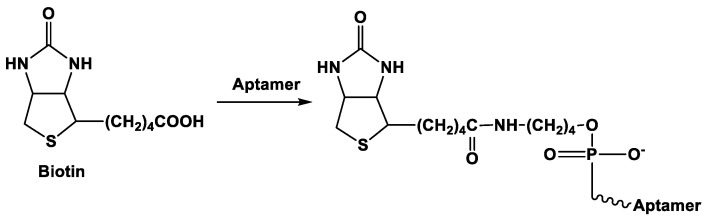
Biotinylated aptamer synthesis.

**Figure 7 sensors-20-06963-f007:**
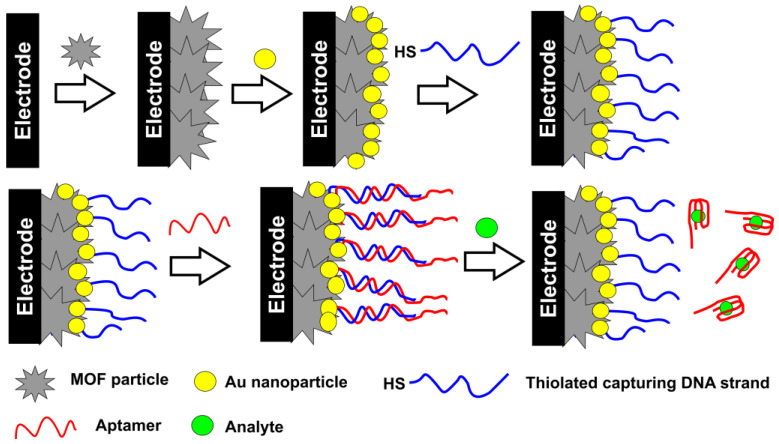
Immobilization of aptamer molecules via hybridization with auxiliary DNA strand and the following interaction with an analyte releasing the aptamer from the surface layer.

**Figure 8 sensors-20-06963-f008:**
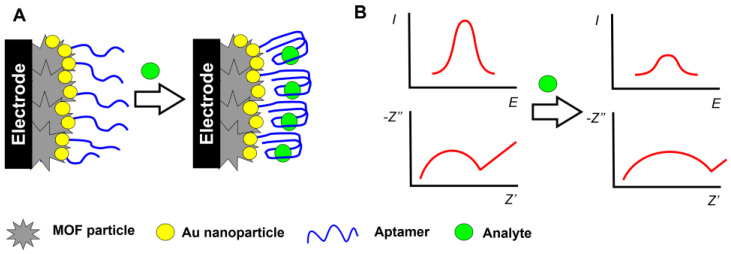
Monitoring aptamer–analyte interactions by the permeability of the surface layer toward small ions. (**A**) Mechanism of interaction; (**B**) changes in the signals recorded with DPV and EIS.

**Figure 9 sensors-20-06963-f009:**
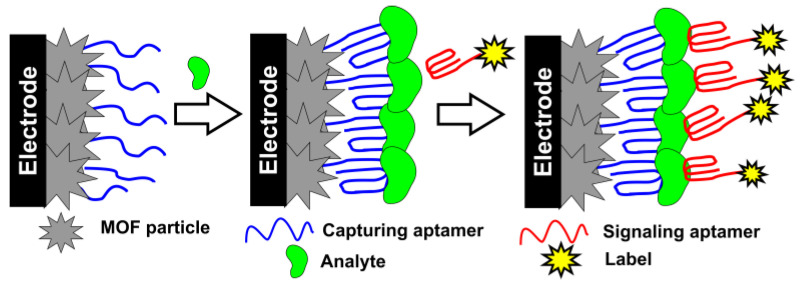
Detection of aptamer–analyte interaction with sandwich analysis using redox-active label.

**Figure 10 sensors-20-06963-f010:**
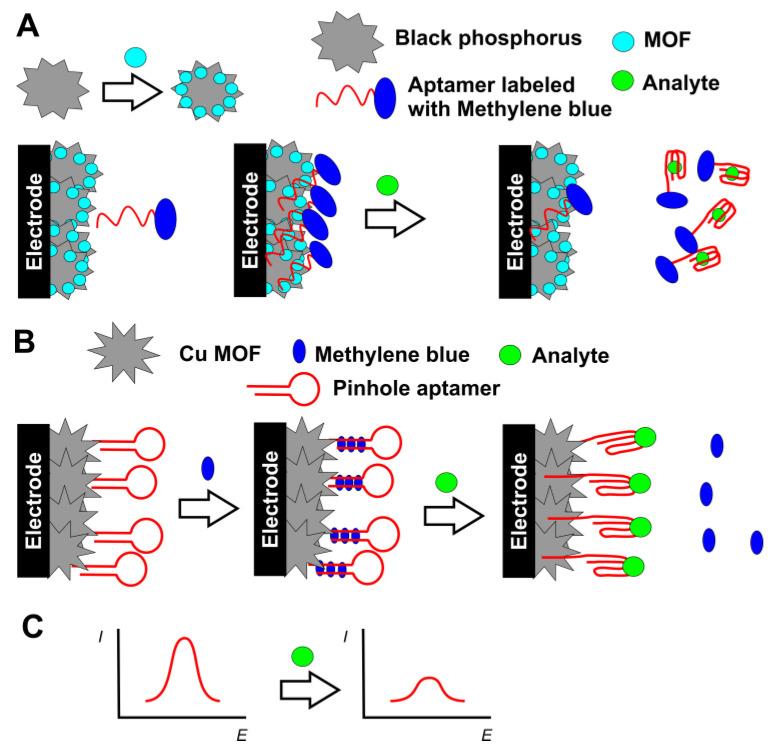
(**A**) Displacement protocol for determination of phosphoprotein 1 with methylene blue-labeled aptamer; (**B**) Determination of *E. coli* with pinhole aptamer saturated with methylene blue; (**C**) Changes in the DPV signal of methylene blue corresponded to the analyte binding.

**Table 1 sensors-20-06963-t001:** The characteristics of electrochemical MOF-based aptasensors. LOD - limit of detection.

Target	MOF Precursors and Synthesis Protocol	Surface Layer Content and Signal Measurement Protocol	Linearity Range/Limit of Detection (LOD)	Ref.
Adenosine	ZnNi MOF from terephtalic acid and metal salts	ZnNi MOF dispersion drop casted on bare Au electrode, aptamer immobilized by electrostatic accumulation, EIS measurements with the [Fe(CN)_6_]^3−/4−^ redox indicator	1×10^−4^–100 ng/mL, LOD 20.32 fg/mL	[64]
Aflatoxin B1	Cu MOF from Cu(NO_3_)_2_ and 2-aminoterephtalic acid followed by conjugation with Au nanoparticles	Bare Au electrode modified with a pinhole DNA, aflatoxin B1 reacts with an aptamer partially hybridized with auxiliary DNA sequence. Releases single-stranded DNA interacts with a pinhole DNA to form double stranded DNA, which is hydrolyzed by DNA exonuclease in a manner to leave the piece of the sequence complementary to the DNA probe bearing Cu MOF with Au nanoparticles. DPV measurements with the [Fe(CN)_6_]^3−/4−^ redox indicator	10^−6^–1 ng/mL, LOD 6.71 × 10^−7^ ng/mL	[104]
Ampicillin	Co MOF on terephtalonitrile covalent organic framework particles from dicyanobenzene, ZnCl_2_ and Co(NO_3_)_2_	Co MOF – terephtalonitrile covalent framework nanosheets deposited on bare Au electrode followed by physical adsorption of aptamer via hydrophobic interactions and hydrogen bonding, EIS measurements with the [Fe(CN)_6_]^3−/4−^ redox indicator	1.0 fg/mL–2.0 ng/mL, LOD 0.217 fg/mL	[105]
Adenosine triphosphate (ATP)	Ce MOF from 2-aminoterephtalic acid and CeCl_3_ in NaOH	Ce MOF dispersion drop casted on bare Au electrode, aptamer immobilized by hydrophobic interactions, hydrogen bonding and electrostatic accumulation, EIS measurements with the [Fe(CN)_6_]^3−/4−^ redox indicator	10 nM–1000 μM, LOD 5.6 nM	[88]
β-Amyloids	Cu MOF from 1,3,5-benzenetricarboxylic acid and CuSO_4_ conjugated with Au nanoflowers	GCE modified with Au nanoparticles bearing thiolated aptamer, Cu MOF-Au nanoflowers conjugate used as label in sandwich assay, DPV signal related to the Cu^2+^ reduction	1 nM–2 μM, LOD 0.45 nM	[106]
β-Amyloids	Ru MOFs from Zn(NO_3_)_2_, Ru(II) bipyridine complex and 2-aminoterephtalic acid dissolved in ethanol and DMF	Aptamer physically adsorbed on the Ru MOF by electrostatic assembling of PVP – nafion layers, GCE modified with g-C_3_N_4_ nanosheets and aptamer-MOF composite. Detection of electrochemical luminescence based on energy transfer mechanism in the presence of S_2_O_8_^2−^ oxidant	10^−5^–500 ng/mL, LOD 3.9 fg/mL	[107]
Carbohydrate antigen CA 125	Tb MOF from TbCl_3_ and 1,3,5-benzenetricarboxylic acid, Fe MOF from FeCl_3_ and terephthalic acid	Bare Au electrode covered with suspension of Tb MOF followed by treatment with suspension of Fe MOF and by physical adsorption of the aptamer, EIS measurements of intrinsic redox activity of the modifier	0.0001–200 U/mL, LOD 5.8 × 10^−5^ U/mL	[108]
C Reactive protein (CRP)	Ni MOF from Ni(NO_3_)_2_ and 2-aminoterephthalic acid in DMF, water or their mixture	Au electrode modified with Ni MOFs with deposited Au nanostar particles. Anti-CRP antibodies and pinhole DNA strands with terminal Methylene blue labels co-immobilized on Au. Target binding is detected by consecutive attachment of the aptamer activating changed in conformation of labeled DNA and its cleavage by DNA exonuclease. As a result, electrochemical conversion of Methylene blue catalyzed by the Ni MOF is detected with square wave voltammetry	1 pg/mL–100 ng/mL, LOD 0.029 pg/mL	[81]
Carcinoembryonic antigen (CEA)	Pb MOF from Pb(NO_3_)_2_ and 2-aminoterephtalic acid modified with capturing single-stranded DNA	Bare Au electrode modified with thiolated aptamer and auxiliary DNA loops to form dendritic DNA scaffold via partially hybridized parts of the sequences by a hybrid chain reaction. After the CEA binding, Pb^2+^ ions from the DNA-MOF tags released are determined by square-wave voltammetry	0.001–100 ng/mL, LOD 0.333 pg/mL	[109]
Carcinoembryonic antigen (CEA)	Zr MOF from ZrCl_4_ and terephtalic acid in DMF	Mixture of AgNO_3_, Zr MOF suspension and aptamer were dissolved in DMF in the presence of HCl and trifluoroacetic acid and reduced by NaBH_4_ to form Ag nanoparticles coupled with aptamer/Zr MOF. They were drop casted on the Au electrode and EIS and DPV measurements performed with the [Fe(CN)_6_]^3−/4−^ redox indicator	0.01–10 ng/mL, LOD 8.88 (EIS) and 4.93 (DPV) pg/mL	[110]
Carcinoembryonic antigen (CEA)	Fe MOF from Fe and 1,4-benzenedicarboxylic acid obtained on the surface of Au – polydopamine nanocomposite	Au@polydopamine@Fe-MOF particles with immobilized aptamer in bovine serum albumin matrix by cross-binding with glutaraldehyde are deposited on bare Au electrode; DPV measurements with the [Fe(CN)_6_]^3−/4−^ redox indicator	0.001–10 000 pg/mL, LOD 0.33 fg/mL	[111]
Cocaine	Zr MOF from ZrCl_4_ and 4′,4″,4″’-nitrilotris([1,1′-biphenyl]-4-carbo- xylic acid)	Bare Au electrode modified with dispersion of Zr MOF with embedded Au nanoparticles and covalently attached thiolated aptamer, EIS and DPV measurements with the [Fe(CN)_6_]^3−/4−^ redox indicator	0.001–1.0 ng/mL, DOI 0.44 (EIS) and 0.75 (DPV) pg/mL	[112]
Colorectal cancer cells CT26	Cr MOF from Cr(NO_3_)_3_ and terephtalic acid on Co phtalocyanine nanoparticles	Bare Au electrode covered with Cr MOF (Cr MOF – Co phtalocyanine hybrid) followed by aptamer physical immobilization, DPV and EIS measurements with the [Fe(CN)_6_]^3−/4−^ redox indicator	50–1 × 10^7^ cells/mL, LOD 38 (EIS) and 8 (DPV) and 31 (EIS) cells/mL	[84]
*Esherichia coli* O157:H7	Cu MOF from Cu(NO_3_)_2_ and 1,3,5-benzenetricarboxylic acid	GCE covered with polyaniline – MOF composited obtained by chemical oxidation of aniline in suspension of the MOF, aptamer immobilized by cross-linking with glutaraldehyde. DPV measurements of the signal of Methylene blue as redox indicator	21–2.1 × 10^7^ CFU/mL, LOD 21 CFU/mL	[113]
HER2 living cancer cells	ZrHf MOFs from ZrCl_4_ and HfCl_4_ mixed with 2-aminoterephtalic and formic acids in DMF, final product of hydrothermal synthesis embedded with carbon nanodots	Bare Au electrode is covered with suspension of the MOFs and then aptamer is physically adsorbed on its surface, DPV and EIS measurements with the [Fe (CN)_6_]^3−/4−^ redox indicator	1 × 10^−4^–10 ng/mL. LOD 30 (EIS) and 19 (DPV) fg/mL	[114]
Insulin	Nanohybrids of Ni/Fe_2_O_3_/NiCo_2_O_4_ obtained by pyrolysis of MOF-on-MOF hierarchical nanostructure from Co (NO_3_)_2_, Ni (NO_3_)_2_ and 2-methylimidazole; solvothermal synthesis and calcination at 600 °C	Bare Au electrode covered with suspension of nanohybrids, aptamer immobilized by physical adsorption; EIS measurements with the [Fe (CN)_6_]^3−/4−^ redox indicator	0.01 pg/mL–100 ng/mL, LOD 9.1 fg/mL	[115]
Kanamycin and chloramphenicol	Zr MOF from ZrCl_4_ and 2-aminoterephtalic acid saturated with Pb^2+^ or Cd^2+^ ions	Magnetic beads bearing capture single stranded DNA complementary to the aptamer attached to the MOF nanoparticles, target interaction releases MOFs, DPV measurement of the Pb^2+^/Cd^2+^ signals	0.02–100 nM, LOD of kanamycin 0.16 pM, of chloramphenicol 0.19 pM	[116]
Kanamycin and oxytetracycline	Zr MOF from ZrCl_4_ and 2-aminoterephtalic acid saturated with Pb^2+^ or Cd^2+^ ions	Magnetic beads bearing capture single stranded DNA complementary to the aptamer attached to the MOF nanoparticles, target interaction releases MOF-aptamer conjugate and RecJf exonuclease amplifies the signal due to formation of free MOF nanoparticles; DPV measurement of the Pb^2+^/Cd^2+^ signals	0.5 pM–50 nM, LOD 0.15 pM (kanamycin) and 0.18 pM (oxytetracycline)	[117]
Lipopolysacharide	Ce MOF from Ce(NO_3_)_3_ and 2-amoinoterephtalic acid	Bare GCE covered with eletrodeposited Au nanoparticles and thiolated capture aptamer. Duplex DNA reacted with an analyte, reporter DNA released in involved in recycling cycle with Zn^2+^ assisted DNAzyme to form the strand complementary to signaling DNA sequence bearing the Au nanoparticles/Ce MOF composite. DPV signal of ascorbic acid electrocatalytic oxidation	10 fg/mL–100 ng/mL, LOD 3.3 fg/mL	[118]
Lipopolysacharide	Cu MOF from Cu(NO_3_)_2_ and 1,3,5-benzenetricarboxylic acid and urea	Bare Au electrode covered with electropolymerized pyrrole and pyrrole-carboxylic acid followed by carbodiimide binding of the aptamer. Cu MOFs are deposited on the analyte - aptamer complex; DPV signal of Cu^2+/+^ ions	1.0 pg/mL–10.0 ng/mL, LOD 0.29 fg/mL	[119]
Lysozyme	Zr MOF from ZrOCl_2_, benzoic or nicotinic acid, acetic acid and 4,4′,4′’- s-triazine-2,4,6-triyltribenzoic acid in DMF	Bare Au electrode modified with the Zr MOF suspension and electrostatically adsorbed aptamer, EIS measurements with the [Fe(CN)_6_]^3-/4-^ redox indicator	0.005–1.0 ng/mL, LOD 3.6 pg/mL	[120]
Michigan cancer foundation cancer cells (MCF-7)	Zr MOFs based on terephtalic, 2-aminoterephthalic acids and the mixture of the 2-aminoterephthalic and 2,5-diaminoterephthalic acid	Bare Au electrode covered with suspension of Zr MOF followed by physical adsorption of the aptamer. EIS measurements with the [Fe(CN)_6_]^3−/4−^ redox indicator	100–10^5^ cells/mL, LOD 31 cells/mL	[91]
microRNA	Cu MOF from Cu(NO_3_)_2_ and 4,4′,4′’,4′’’-(porphine-5,10,15,20-tetrayl)- tetrakisbenzoic acid doped with thionine	GCE covered with suspension of black phosphorus nanosheets mixed with Cu MOFs, nafion and thionine followed by adsorption of ferrocene labeled aptamer; DPV signal of thionine and ferrocene	2 pM–2 μM, LOD 0.3 pM	[121]
microRNA	Fe MOF from FeCl_3_ and 2-aminoterephtalic and acetic acid in DMF; After synthesis, Pd nanoparticles were deposited from the suspension of the Fe MOFs in PEI	GCE modified with g-C_3_N_4_ and Au nanoparticles followed by immobilization of capture probe. After miRNA binding, Fe MOFs particles with the signaling aptamer attached via avidin-biotin binding were added. Electrocatalytic signal of tetramethylbenzidine in the presence of hydrogen peroxide recorded by amperometry and EIS response with the [Fe(CN)_6_]^3−/4−^ redox indicator	10 fM to 100 nM, LOD 10 fM (EIS) 0.01 fM – 10 pM, LOD 0.003 fM (amperometry)	[122]
Mucin 1	Cu MOF from Cu(NO_3_)_2_ and 1,3,5-benzenetricarboxylic acid in aqueous DMF	GCE covered with suspension of Cu MOF and reduced graphene oxide, aminated aptamer immobilized by carbodiimide binding, DPV signal of Cu^2+/+^ ions	0.1 pM–10 nM, LOD 0.033 pM	[123]
Mucin 1	Zr MOF containing Zr_12_ clusters from ZrCl_4_ and 4,4′-biphenyldicarboxylate doped with Ru bipyridine and 2,2′-bipyridine-5,5′- dicarboxylic acid complex in DMF and formic acid	GCE covered with Ru MOF dispersed in chitosan and with Pt nanoparticles. Then, two auxiliary hairpin probes were added to form self-assembled DNA in the presence of the analyte molecules. The product is digested with *exo* III DNAze and the piece of the aptamer left attached on the electrode formed G4 quadruplex with hemin. Finally, electrochemiluminescence was measured	1–10^7^ fg/mL, LOD 0.14 fg/mL	[124]
Mucin 1	2D Zr MOF from ZrOCl_2_, trifluoroacetic acid and 4’,4’’’,4’’’’-nitrilotris [1,1’-biphenyl]- 4-carboxylic acid in diethylfromamide	Bare Au electrode covered with suspension of the MOF nanosheets, EIS measurements with the [Fe(CN)_6_]^3−/4−^ redox indicator	0.001–0.5 ng/mL, LOD 0.12 pg/mL	[125]
Mycobacterium tuberculosis antigen	Fe MOF from FeCl_3_ and 2-aminoterephtalic acid in DMF covered with PEI	GCE covered with Au-Pt bimetallic nanoparticles and Fe MOF, capture aptamer adsorbed on the modifier layer. Signaling aptamer attached to the particles consisted of *N*-doped carbon nanotubes, reduced graphene oxide, fullerene and Au nanoparticles. Sandwich assay, DPV signal of fullerene	1–10^6^ fg/mL, LOD 0.33 fg/mL	[126]
Mycobacterium tuberculosis antigen	Au containing Zr MOF from ZrCl_4_, 2-aminoterephtalic acid, HAuCl_4_ and PVP in DMF-H_2_O	Bare Au electrode modified with thiolated capturing aptamer. Sandwich assay with signaling aptamer attached to the MOF particles together with horseradish peroxidase, DPV signal of hydroquinone/benzoquinone as enzyme substrate/product	0.02−1000 pg/mL, LOD 10 fg/mL	[127]
Mycobacterium tuberculosis antigen	Fe MOF from FeCl_3_ and 2-fminoterephtalic acid in DMF	GCE covered with composite consisted of reduced graphene oxide, MOF, toluidine blue, Au@Pt nanoparticles and PDDA. Direct current voltammetry of toluidine blue	1.0 × 10^−4^–2.0 × 10^2^ ng/mL, LOD 3.3 × 10^−5^ ng/mL	[128]
Ochratoxin A	Cd MOF from Cd acetate and 2,5-dihydroxyterephtalic acid in DMF	GCE covered with Au nanoparticles and MoS_2_ in chitosan matrix, auxiliary DNA complementary to aptamer immobilized by physical adsorption and then hybridized with an aptamer. Target binding release aptamer from the electrodeposited layer; DPV signal of Cd from the MOF	0.05–100 ng/mL, LOD 10 pg/mL	[129]
Oxytetracycline	Ce MOF from Ce(NO_3_)_3_ and 1,3,5-benzenetricarboxylic acid in ethanol/water	Melamine and cyanuric acid were dissolved in dimethylsulfoxide in 1:1 ratio and mixed with Ce MOF to obtain aggregates deposited on the Au electrode; aptamer immobilized on the modifier layer. EIS measurements with the [Fe(CN)_6_]^3−/4−^ redox indicator.	0.1−0.5 ng/mL, LOD 17.4 fg/mL	[130]
Oxytetracycline	Fe(II) MOF from FeSO_4_ and 4,4′,4′′-nitrilotrisbenzoic acid in *N*-methyl-2-pyrrolidone and *N*-methylformamide	The Fe(II) MOF particles pyrolyzed at 550 °C, aptamer adsorbed on their surface. EIS measurements with the [Fe(CN)_6_]^3−/4−^ redox indicator	0.005−1.0 ng/mL, LOD 0.027 pg/mL	[87]
Patulin	Cu MOF from Cu(NO_3_)_2_ and 2-aminoterephtalic acid in PVP and DMF	GCE first modified with *N*-doped graphene quantum dots. Then, Cu MOFs were decorated with Au nanoparticles by chemical reduction of HAuCl_4_ with NaBH_4_ and placed on the electrode. Finally, aniline was polymerized in the presence of patulin as template. EIS measurements with the [Fe(CN)_6_]^3-/4-^ redox indicator	0.001 to 70.0 ng/mL, LOD 7 × 10^−4^ ng/mL	[131]
Patulin	Zr MOF from ZrCl_4_, 2-aminoterephtalic acid and dodecanoic acid in DMF	Zr MOF saturated with Methylene blue was mixed with glutaraldehyde and incubated with aminated aptamer and placed on the Au electrode modified with ZrO nanoparticles in chitosan matrix; DPV signal of Methylene blue as redox indicator	5×10^−8^–0.5 μg/mL, 1.46 × 10^−8^ μg/mL	[132]
Pb^2+^	Fe MOF from FeCl_3_ and terephtalic acid in DMF	Fe MOF decorated with AgPt nanoparticles obtained from H_2_PtCl_6_ and AgNO_3_ and ascorbic acid; Fe MOPFs modified with polyethylene imine and AgPt nanoparticles in chitosan presence and covered with bovine serum albumin on the GCE; electrocatalytic response to H_2_O_2_ measured by DPV	0.1 pM–100 nM, LOD 0.032 pM	[133]
Pb^2+^ and As^3+^	Fe MOF – Fe_3_O_4_@C double shell nanocapsules obtained by mixing FeCl_3_ and 2-amionoterephtalic acid in the presence of Fe_3_O_4_@C suspension	Bare Au electrode covered with suspension of composite nanoparticles followed by physical adsorption of the aptamers toward Pb^2+^ and As^3+^; EIS measurements with the [Fe(CN)_6_]^3-/4-^ redox indicator	Pb^2+^: 0.01–10.0 nM, LOD 2.27 pM; As^3+^: 0.01−10.0 nM, LOD 6.73 pM	[134]
Penicillamine	Co MOF from Co(NO_3_)_3_, 2-methylimidazole and *N,N*-hexadecanedioyldi-L-glutamic acid in methanol	Bare GCE covered with the Co MOF based enantioselector, Direct current voltammetry	3.25−19.50 mM in racemic mixture	[135]
Protein tyrosine kinase-7 (PTK-7)	Zr MOF on Zn MOF structures from Zn(NO_3_)_2_, ZrOCl_2_, 2-methylimidazole in trifluoroacetic acid and diethylformamide	ZnZr MOF nanoparticles deposited on the GCE, EIS and DPV measurements with the [Fe(CN)_6_]^3−/4−^ redox indicator	0.001–1 ng/mL (EIS and DPV), LOD 0.84 (EIS), 0.66 (DPV) pg/mL	[136]
*Salmonella typhimurium*	Zr MOF from ZrCl_4_, biphenyl-4,4′-dicarboxylic acid in acetic acid and DMF in the presence of graphene	GCE covered with Zr MOF-graphene composite and auxiliary DNA sequence complementary to aptamer. Competitive assay with aptamer labeled with Au nanoparticles/horseradish peroxidase conjugate. DPV measurements of hydroquinone signal as enzyme substrate	20–2 × 10^8^ cfu/mL, LOD 5 cfu/mL	[137]
Stress induced phosphoprotein 1	Mn MOF from Mn(NO_3_)_2_, Ni(NO_3_)_2_, terephtalic acid in DMP	Black phosphorus dispersed in *N*-methyl-2-pyrrolidinon and added to the Mn MOF in ethanol. The mixture was dispersed on the GCE surface followed by the adsorption of the aptamer bearing methylene blue, DPV measurement of the Methylene blue signal as redox label	2×10^−3^–1 × 10^4^ ng/mL, LOD 1 pg/mL	[138]
Thrombin	Cu MOF from Cu(NO_3_)_2_ and *trans*-1,4- cyclohexanedicarboxylic acid in DMF	GCE covered with the Cu MOF and electrodeposited Au nanoparticles followed by immobilization of thiolated aptamer; direct current voltammetry of Cu^2+/+^ redox pair	0.01 fM–10 nM, LOD 0.01 fM	[139]
Thrombin	Ni MOF from Ni(NO_3_)_2_ and 4,4′,4′’-tricarboxytriphenylamine	GCE modified with Au nanoparticles obtained by cathodic reduction of HAuCl_4_ followed by immobilization of thiolated capture aptamer. Sandwich assay with Ni MOF decorated with Au nanoparticles and signaling aptamer; DPV signal of Ni MOF	0.05 pM–50 nM, LOD 0.016 pM	[140]
Thrombin	Fe MOF from FeCl_3_ and 2-aminoterephtalic acid in acetic acid and DMF	Bare Au electrode modified with Pt nanoparticles and thrombin. Au electrode covered with complementary DNA sequence hybridized with the aptamer. Reaction with thrombin releases aptamer-thrombin complex able to catalyze oxidation of tetramethylbenzidine in the presence of H_2_O_2_ (colorimetric signal). Simultaneously, remained capture DNA reacts with signaling DNA bearing Au nanoparticles. DPV signal of Methylene blue as redox indicator	1 fM 10 nM (DPV), 0.5 pM−1 nM (colorimetry), LOD 0.33 fM (DPV) and 0.17 pM (colorimetry)	[141]
Thrombin	Ce MOF from Ce(NO_3_)_3_ and 1,3,5-benzenetricarboxylic acid with partial oxidation to Ce(IV) with H_2_O_2_	GCE covered with electrodeposited Au nanoparticles with thiolated capture pinhole aptamer. Thrombin initiates exonuclease III assisted recycling amplification of the DNA strands complementary to the signaling DNA attached to Ce MOF bearing Au nanoparticles and thionine as redox indicator. Square wave voltammetry of thionine mediated by Ce(III)/Ce(IV) redox pair of the MOF	0.1 fM–10 nM, LOD 0.06 fM	[142]
Thrombin	Co MOF from CoSO_4_, 2-aminoterephtalic acid in DMF and ethanol	GCE covered with electrodeposited Au nanoparticles and thiolated aptamer. Sandwich assay with signaling aptamer attached to the Co MOF decorated with Pt nanoparticles, DPV signal of Co(II)/Co(III) redox pair of the MOF	0.1 pM–50 nM, LOD 0.33 fM	[143]
	Fe MOF from FeCl_3_ and 2-aminoterephtalic acid in acetic acid and DMF	GCE covered with electrodeposited Au nanoparticles and thiolated capture aptamer. Sandwich assay with signaling aptamer attached to the Fe MOF modified with Au nanoparticles, hemin and glucose oxidase. DPV signal of enzymatic glucose oxidation	0.0001–30 nM, LOD 0.068 pM	[144]
Thrombin	Co MOF from Co(NO_3_)_3_ and 2-aminoterephtalic acid	Bare GCE covered with electrodeposited Au nanoparticles and pinhole auxiliary DNA involved in d target-triggering nicking enzyme signaling amplification with two other pinhole DNA probes followed by attachment of the Co MOF bearing Pt-Pd nanoparticles. DPV signal of electrocatalytic oxidation of H_2_O_2_	1 pM–30 nM, LOD 0.32 pM	[145]
Thrombospondin-1	Ce MOF from Ce(NO_3_)_3_ and 1,3,5-benzenetricarboxylic acid followed by partial oxidation with H_2_O_2_/NaOH	Ce MOF particles were decorated with HAuCl_4_/NaBH_4_ in the presence of PVP and DNA-aptamer hybrid bearing Au/Pt/Ru nanoparticles. GCE covered with the Ce MOF nanoparticles with Au nanoparticles. Reaction of the analyte with DNA-aptamer performed prior to reaction with electrode where hybridization of signaling probe with complementary sequence takes place. DPV current of the Ce^3+/4+^ measurements	1 fg/mL–10 ng/mL, LOD 0.13 fg/mL	[146]
Tobramycin	Ce/Cu MOF from Ce(NO_3_)_3_, Cu(NO_3_)_2_, 1,3,5-benzenetricarboxylic acid in ethanol pyrolized at 900 °C	Bare Au electrode covered with the Ce/Cu MOF suspension, aptamer physically adsorbed on the modifier layer; EIS measurements with the [Fe(CN)_6_]^3−/4−^ redox indicator	0.01 pg/mL–10 ng/mL, LOD 2.0 fg/mL	[147]
Vibrio parahaemolyticus	Ru MOF from tris(4,4′-dicarboxylicacid-2,2′- bipyridyl)ruthenium(II) dichloride, Zn(NO_3_)_2_ in *n*-propanol	Bare Au electrode covered with thiolated capture aptamer, Ru MOF saturated with Pb^2+^ ions and signaling aptamer via carbodiimide binding. Sandwich assay with electroluminescent Ru signal and DPV Pb signal	5–1 × 10^8^ cfu/mL (ECL), 0.1 × 10^8^ cfu/mL (DPV), LOD 1.7 (ECL), 3.3 (DPV) cfu/mL	[148]
Zearalenone	Co MOF decorated with PtNi nanoparticles from Ni(NO_3_)_2_, K_2_PtCl_6_, CoCl_2_, 2-aminoterephtalic acid in DMF	Bare Au electrode was consecutively treated with CoSe/Au nanorods and auxiliary DNA partially hybridized with capture aptamer – zearalenone complex. The hybridization complex was then cleaved and finally interacted with thionine saturated Co MOF bearing complementary signaling DNA, DPV signal of thionine as redox indicator	10.0 fg/mL–10.0 ng/m, LOD 1.37 fg/mL	[149]

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
