# Peer review of "Electrochemical Aptasensors Based on Hybrid Metal-Organic Frameworks"

_sensors, 2020, doi:10.3390/s20236963_

Round 1

Reviewer 1 Report

In the present review, the application of the MOFs in electrochemical aptasensors are briefly discussed, paying special attention to the role of the MOF materials in the aptamer immobilization and signal generation. In my opinion, the review is an excellent work summarizing the application of MOFs and its utilities in the world of aptasensors. The review is very well written, is easy to read and contains a great amount of information, as corresponds to a Review article. In my mind, it should be published almost in its present form, but the Conclusions section should be rewritten to focus more on the applications of MOFs to aptasensors, as mentioned in the title of the article.

Minor English corrections:

- Correct the spelling of voltammetric in line 180 and aminoterephtalic acid in line 239 (Fig 3 legend).

- Line 486.- Replace figure 6 by figure 9.

Sincerely,

The reviewer

NOTE: Make sure that you are the owner of the copyright or have the permission to reprint the image.

Author Response

We are grateful to this reviewer for very useful comments that allowed us to improve manuscript

Comment: The Conclusions section should be rewritten to focus more on the applications of MOFs to aptasensors, as mentioned in the title of the article.

Response: The following changes in Conclusion have been made:

Second paragraph was added:

All the above mentioned is applicable for electrochemical aptasensors utilizing MOFs to enhance surface square, immobilize aptamers and generate redox signals. The use of bulky and electrochemically inactive aptamer molecules makes the aptasensor behavior sensitive toward the conditions of electron transfer. Contrary to electroconductive carbonaceous materials, MOF particles offer shuttle mechanism of the electron exchange with participation of both free and immobilized mediators including the metals as counterparts of the MOF structure. Such an electron transfer path is easily braked by inclusion of analyte molecules. For this reason, aptamers based on the MOF structures often show record sensitivity of detection especially for small analyte molecules that affect the transfer of conventional redox indicators like [Fe(CN)6]3-/4- to rather small extent if common approaches of the biosensor assembling are used.

Third paragraph was added:

Although most of the articles devoted to the electrochemical aptasensors based on MOF materials do not provide comprehensive comparison of the characteristics with conventional analogs, it can be concluded that introduction of MOF particles can decrease the LOD values by at least one order of magnitude against application of similar heterogeneous mediators of electron transfer. The difference might be even higher if MOF particles are used as parts of indicators attached to the electrode interface via target biochemical interactions with analytes. These benefits follow from the size of the particles and multiplication of the signal due to much higher number of redox sites involved in the electron transfer. Appropriate examples have been presented in Table 1. It should be also noted that other figures of merit like dynamic range of concentration or measurement duration did not demonstrate similar progress against other biosensor materials. Then, application of biochemical approaches to the signal increase including implementation of auxiliary pinhole aptamers and exonuclease assisted DNA amplification level the influence of other components of the aptasensor assembly.

Sixth and seventh paragraphs were modified:

The following trends in the further progress of MOF based aptasensors can be mentioned.

  1. Faster progress is expected in multiplex assay and simultaneous determination of several analytes based on a number of aptamers introduced in the aptasensor structure.
  2. Growing interest will be directed to the signal-on aptasensors, where increased concentration of an analyte increases their response. This kind of sensors seems more beneficial both from the point of view of metrological assessment of the results and application of the aptasensors in real sample assay.
  3. Although the term ‘MOF’ is mostly related to the use of the linkers with carboxylic binding groups, the variety of organic part of the MOFs will be increased by introduction of new structures and precursors derived from imidazole fragments and some others exerting unusual architecture of internal space of the particles and their morphology.
  4. The interest to the synthesis of new MOF materials will be shifted to their implementation in supramolecular structures consisting of supramolecular polymers and aptamer molecules.

In general, high variability of the MOF structures and their easy accessibility as building blocks of aptasensors make it possible to rely on their broad application in in medicine, food industry and environmental pollution monitoring.

Comment: Minor English corrections:

- Correct the spelling of voltammetric in line 180 and aminoterephtalic acid in line 239 (Fig 3 legend).

- Line 486.- Replace figure 6 by figure 9.

Response: Changes required have been made

Comment: Make sure that you are the owner of the copyright or have the permission to reprint the image.

Response: The image presented has been taken from the Open Access journal and was correctly referred to. Other figures were prepared by the authors in one style using CorelDraw software.

Reviewer 2 Report

The progress in the synthesis and application of the MOFs in electrochemical aptasensors were summarized and discussed in this review. The authors prepared a well-organized review. I suggest acceptance after minor review.

1) If possible, add some discussion about the enhancement of the detection sensitivity with the MOFs. “For example, XXX fold could be achieved after using the MOFs.”

2) Some statistical analysis should be added. For example, a pie chart can be used to show the readers how many studies used the MOFs for aptamer immobilization and how many for signal generation, and so on.

Author Response

We are grateful to this reviewer for very useful comments that allowed us to improve manuscript

Comment: 1) If possible, add some discussion about the enhancement of the detection sensitivity with the MOFs. “For example, XXX fold could be achieved after using the MOFs.”

Response: The following paragraphs were added into the Conclusion:

Sixth and seventh paragraphs:

Although most of the articles devoted to the electrochemical aptasensors based on MOF materials do not provide comprehensive comparison of the characteristics with conventional analogs, it can be concluded that introduction of MOF particles can decrease the LOD values by at least one order of magnitude against application of similar heterogeneous mediators of electron transfer. The difference might be even higher if MOF particles are used as parts of indicators attached to the electrode interface via target biochemical interactions with analytes. These benefits follow from the size of the particles and multiplication of the signal due to much higher number of redox sites involved in the electron transfer. Appropriate examples have been presented in Table 1. It should be also noted that other figures of merit like dynamic range of concentration or measurement duration did not demonstrate similar progress against other biosensor materials. Then, application of biochemical approaches to the signal increase including implementation of auxiliary pinhole aptamers and exonuclease assisted DNA amplification level the influence of other components of the aptasensor aassembly

Comment: 2) Some statistical analysis should be added. For example, a pie chart can be used to show the readers how many studies used the MOFs for aptamer immobilization and how many for signal generation, and so on.

The following text was added into a new Section 3.1:

3.1. General assessment and bibliography statistics

Biosensor assembling is a protocol of consecutive attachment of biochemical and auxiliary components on the transducer interface to reach optimal conditions for target biochemical interactions and signal generation. Regarding electrochemical biosensors, heterogeneous mediators of electron transfer are mostly required to compensate for decreased conductivity of the surface layer caused by implementation of biopolymers and to amplify the signal of redox active labels and indicators. Immobilization of biomolecules should maintain their microenvironment most to be comfortable for biochemical functioning and protect them from reactive species and undesired working conditions (extreme pH, influence of reactive species on the steps of biosensor assembling and signal measurement etc.). Besides, immobilization should take into account steric limitations of the analyte binding related to the partial shielding of biochemical receptors on the electrode interface. In the history of biosensors, many approaches have been elaborated for biosensor assembling that offer stable signal and long-term operation of biosensors in different and changing conditions. Below, some of them are briefly considered in relation to the use of the MOFs in aptasensors. It should be clearly noted that the protocols described below do not exhaust all the methods and are limited with those most frequently used within the subject of the review.

Although MOFs belong to rather new materials that appeared in the scope of chemists in the past few decades, they have found intensive application in electrochemical sensors and biosensors from very beginning. Figure 4 represents the results of the bibliography statistics performed with Web of Science to follow annual changes in the number of publications and general principles of the MOF application in the aptasensor assembly.

Starting from 2012, an exponential growth of the articles devoted to the use of electrochemistry tools in the MOF synthesis and application was observed. Meanwhile, a significant part of such works utilized redox activity of the metals in the assembly of the MOF particles as a way to assess the structural specificity of the materials and to monitor the assembling of reticular structures from precursors, especially in 2D nets on the solid supports. Then, electrocatalytic properties of the metal clusters in the knots of the frameworks were successfully applied in the protocols of mediated oxidation of many organic species attractive from the point of view of medical diagnostics or food safety control. Application of the MOFs in aptasensors covers about 10-12% of the publications and among them about half is devoted to electrochemical aptasensors. As could be seen from histograms, enormous growth of the interest to the MOF based aptasensors is expected in the nearest future. It is related to remarkable variety of their properties and obvious advantages briefly mentioned above in the Introduction. The role of the MOFs in the aptasensors described (Fig. 4D) varies from rather traditional application as surfaced enhancer and to their use as a source of redox mediators collected or released in the target binding event. MOFs are also well compatible with biochemical strategies of the signal amplification assuming the use of enzymatic and DNAzyme amplification or exonuclease assisted DNA amplification protocols. To some extent, sandwich assay with two aptamer/DNA sequences utilized for immobilization of bioreceptor and its labeling can be considered as biochemical approach previously elaborated for immunoassay. All of the assemblies combining MOFs and aptamers with electrochemical transducer are considered in more detail below in the review.

Reviewer 3 Report

In this paper, the authors summarized the progress in the synthesis and application of Metal-organic Frameworks (MOF) in electrochemical aptasensors. Overall, the manuscript is well organized with the thorough literature reviews. Therefore, I suggest the acceptance of this manuscript after accommodating the following comments.

  • It would be better to include the brief explanation for not only the general synthesis process of MOF, but also MOF based on the type of metal ions.
  • There are many typos and grammatical mistakes that should be carefully checked and corrected. (One example) they can serve as specific sorbents for redox active species which activity is then used for the signal measurement <”which activity” is grammatically wrong>. This is just one example. The authors need to get their manuscript edited by the native speakers.
  • More works about MOF and aptasensors can be cited in the introduction. For example, org/10.1016/j.bios.2019.111451, doi.org/10.1016/j.bios.2020.112604, doi.org/10.1016/j.bios.2016.10.042, doi.org/10.1016/j.bios.2017.11.028

Author Response

We are grateful to this reviewer for very useful comments that allowed us to improve manuscript

Comment: It would be better to include the brief explanation for not only the general synthesis process of MOF, but also MOF based on the type of metal ions.

Response: The following text has been added in the Introduction:

Most common MOFs described in the assembly of electrochemical sensors and especially biosensors are constructed from carboxylate based linkers. Their stability toward hydrolysis depends on the strength of the bonds between the carboxylate groups and the metal-based nodes. From the hard/soft acid/base principle, hard carboxylates form more robust structures with the metal ions in a high valence state (Ce(III/IV), Zr(IV), Hf(IV) etc.) [56-59]. MOFs assembled from Ni, Cu and Co based nodes and carboxylate bearing linkers degrade to appropriate oxides in basic media [59] For the same reason, such MOFs can be used for the synthesis of derived materials with electrocatalytic properties.  Among the MOFs based on other metals, many reported examples refer to UiO-66 consisted of hexa-zirconium nodes and linear dicarboxylate linkers [60, 61]. They exert high chemical stability both in acidic and neutral media but are redox inactive and hence should be first labeled with some appropriate groups like porphyrin fragments [62]. In aptasensors, high affinity of Zr oxides toward phosphate groups in the DNA sequence make such materials attractive in immobilization of aptamers with no respect of the protocol of signal measurement. MOFs based on nitrogen containing linkers are considered as alternative to the described carboxylate-based MOFs. ZIF (zeolitic imidazolate frameworks) family is most popular in electrochemical detection systems. They are assembled with Zn2+ and Co2+ cations and are quite stable even in basic media due to formation of rather narrow and hydrophobic pores [63].

Comment: There are many typos and grammatical mistakes that should be carefully checked and corrected. (One example) they can serve as specific sorbents for redox active species which activity is then used for the signal measurement <”which activity” is grammatically wrong>. This is just one example. The authors need to get their manuscript edited by the native speakers.

Response: The manuscript was checked by native English speaker

Comment: More works about MOF and aptasensors can be cited in the introduction. For example, 10.1016/j.bios.2019.111451, doi.org/10.1016/j.bios.2020.112604, doi.org/10.1016/j.bios.2016.10.042, doi.org/10.1016/j.bios.2017.11.028

Response: The reference kindly provided by Reviewer have been added to the Introduction section.

Reviewer 4 Report

Submitted manuscript provides a great insight for MOF for developing electrochemical aptasensors. This review article is well supported through multiple data, schematics and comparative table for previously reported literatures. This manuscript may accepted in current format for publication in Sensors.

Author Response

Comment: Submitted manuscript provides a great insight for MOF for developing electrochemical aptasensors. This review article is well supported through multiple data, schematics and comparative table for previously reported literatures. This manuscript may accepted in current format for publication in Sensors.

Response: We are grateful to the reviewer for very positive opinion.